# Beyond Static Snapshots: Physically-Based Deformable and Relightable 2D Gaussians

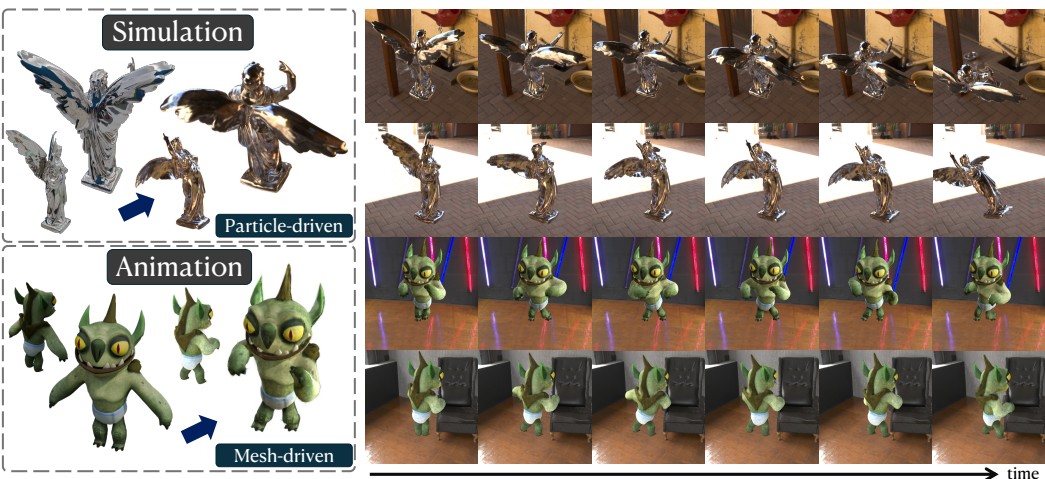

Figure 1: **DR-GS** achieves physically plausible rendering with dynamic deformation and environmental lighting adaptation through decoupled geometry-illumination-material representations.

## Abstract

Gaussian splatting (GS), as a cutting-edge technique in 3D reconstruction, has garnered significant attention in VR/AR and digital content creation due to its explicit parameterization and efficient rendering capabilities. However, existing GS-based methods for deformable objects face two key limitations: **(i)** illumination is erroneously baked into textures, causing physically inconsistent responses under dynamic deformations and lighting changes; **(ii)** snapshots-based reconstruction restricts post-reconstruction material editing. To address these challenges, we propose **D**eformable and **R**elightable **GS** (**DR-GS**), a unified Gaussian framework that integrates physically-based inverse rendering, relighting, and deformation-aware manipulation. Through explicitly disentangling geometry, illumination, and material representations, DR-GS overcomes the limitations of static snapshots, resolving unrealistic appearance under varying conditions while enabling post-reconstruction parameter editing. Experimental results show that DR-GS not only attains state-of-the-art visual fidelity, with particular strength on glossy surfaces, but also creates a fully decoupled geometry-illumination-material pipeline, which enables both high-quality 3D asset creation and comprehensive editing.

## 1 Introduction

Gaussian splatting (GS) has emerged as a novel 3D scene representation method that employs learnable anisotropic Gaussian kernels for efficient explicit geometric modeling. Compared to implicit Neural Radiance Fields (NeRF) (Mildenhall et al., 2020), GS demonstrates superior rendering efficiency and real-time dynamic scene interaction capabilities. These advantages make GS particularly valuable for applications demanding both photorealism and real-time performance, including virtual/augmented reality (VR/AR) (Jiang et al., 2024; Luo et al., 2025; Guo et al., 2025; Li et al.,

2025b), avatars (Li et al., 2025a; Hu et al., 2024; Qian et al., 2024), and embodied AI systems (Lu et al., 2024; Shorinwa et al., 2024; Ji et al., 2024; Zheng et al., 2024; Yu et al., 2025). The explicit nature of GS representation ensures high visual fidelity while achieving remarkable computational efficiency, establishing a new paradigm for real-time 3D reconstruction and interaction.

Current GS-based deformable object modeling primarily adopts a "reconstruct-then-drive" paradigm (Xie et al., 2024; Gao et al., 2025; 2024b; Guédon & Lepetit, 2024; Liu et al., 2024; Huang et al., 2025), where static 3D reconstruction from multi-view images is followed by geometric transformation of Gaussians via intermediate representations such as particle systems or mesh deformations. However, existing methods suffer from two key limitations: (i) illumination is baked directly into Gaussian texture representations, causing physically inconsistent responses under deformation or lighting changes, especially problematic for specular surfaces; (ii) static reconstruction inherently restricts post-reconstruction editing of material properties. Although inverse rendering can disentangle illumination, geometry, and materials for photorealistic relighting, it remains limited to static scenes, leaving dynamic settings unaddressed.

To overcome these challenges, we propose **DR-GS**, an innovative unified deformable Gaussian framework that achieves the first efficient co-processing of inverse rendering, relighting, physics simulation, and 3D animation through flexible representations and shared computational architecture. The key innovations include: (i) comprehensive material modeling spanning diffuse to specular surfaces for static reconstruction; (ii) a novel hybrid driving mechanism supporting both particle systems and mesh deformations for dynamic control. Inspired by GSP (Feng et al., 2025), DR-GS separates simulation and rendering objects while dynamically updating Gaussians via generalized moving least squares (GMLS) (Martin et al., 2010) interpolation.

For rendering challenges in dynamic scenes, DR-GS employs low-sample Monte Carlo estimation of the complete rendering equation, enhanced by multiple importance sampling (MIS), combining cosine-weighted, GGX (Heitz, 2018), and environment light distributions (Pharr & Humphreys, 2004), and cross-bilateral filtering (Schied et al., 2017) for noise reduction. To accelerate rendering, DR-GS stores decoupled material parameters on truncated signed distance fusion (TSDF)-extracted mesh vertices and implementing mesh-based ray tracing, replacing 2D Gaussian ray tracing (Gu et al., 2025a) during continuous deformation under varying illumination. This approach maintains high-quality inter-reflectance and shadow modeling while substantially improving computational efficiency. To summarize, our contributions include:

- **Unified Gaussian framework**: We propose DR-GS, a unified Gaussian representation that supports physically-based inverse rendering, relighting, and deformation-driven control, overcoming limitations of existing in handling geometric deformation and environment illumination response.
- **Efficient dynamic rendering**: We optimize rendering performance in dynamic scenes through MIS and Monte Carlo estimation, accelerating rendering while maintaining high visual realism.
- **Editable parameter pipeline**: Our decoupled geometry-illumination-material pipeline enables high-fidelity 3D asset creation and editing, advancing virtual content creation and simulation.

## 2 RELATED WORKS

**Dynamic Gaussian splatting** related research primarily follows two technical paradigms: learning-based approaches typically take video sequences as input and and incorporate temporal variations features into Gaussians (Yang et al., 2024b; Chen et al., 2023; Wu et al., 2024; Li et al., 2024; Qian et al., 2024; Hu et al., 2024; Yang et al., 2024a; Li et al., 2025a; Ma et al., 2025), while deformation-driven methods adopt a "reconstruct-then-drive" strategy that first performs static reconstruction from multi-view inputs before driving Gaussian deformation through physical simulation or mesh manipulation. Our framework adopts the latter paradigm. Physics-inspired techniques employ material-aware simulation (Cao et al., 2024; Zhang et al., 2024; Huang et al., 2025; Liu et al., 2024; Zhao et al., 2025; Qiu et al., 2024; Luo et al., 2025), such as Material Point Method in PhysGaussian (Xie et al., 2024) and Position-Based Dynamics in GSP (Feng et al., 2025), while geometry-driven methods utilize mesh edition for artist-controlled deformation (Gao et al., 2025; 2024b; Guédon & Lepetit, 2024; Waczyńska et al., 2024; Jiang et al., 2024). Existing methods demonstrate competent static object rendering but fail to handle appearance variations caused by geometric deformation or environmental changes, leading to conspicuous visual artifacts that are especially evident on glossy surfaces. Our DR-GS resolves this critical issue through a decoupled rep-

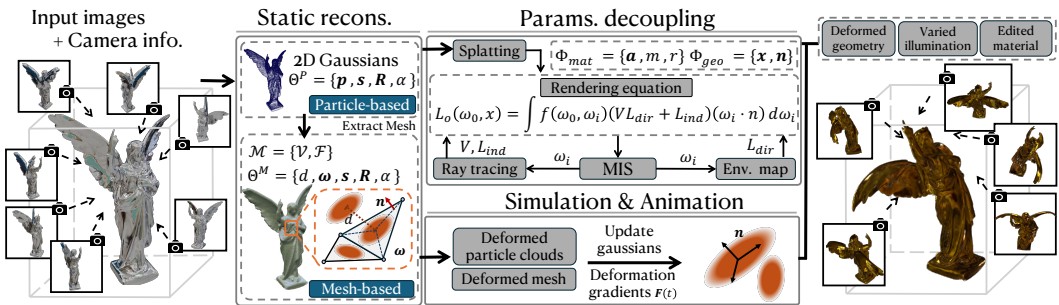

Figure 2: **Overview of *DR-GS*.** Our framework consists of three stages: (i) Static reconstruction: building initial 2D Gaussians and mesh from calibrated images, with optional mesh-based reinitialization; (ii) Parameters decoupling: generating material (*albedo, metallic, roughness*) and geometry (*position, normal*) maps via splatting while solving the rendering equation through MIS and ray tracing; (iii) Dynamic driving: updating Gaussians via deformation gradients between canonical and deformed particle clouds/meshes from physical simulation/animation. Overall, our decoupled geometry-illumination-material framework supports physically plausible rendering under deformations, illumination changes, and material edits with interactive manipulation.

resentation that independently stores material attributes in Gaussians and utilizes physically-based rendering to substantially improve realism under dynamic scenarios, particularly for glossy surfaces.

**Physically-based inverse rendering** seeks to recover geometry, materials, and lighting from images, facing inherent ambiguities between observations and physical parameters. Current methods use differentiable rendering with physics-based light transport: NeRF-based techniques (Srinivasan et al., 2021; Boss et al., 2021; Zhang et al., 2023; Boss et al., 2022; Liu et al., 2023; Gu et al., 2025b; Zhu et al., 2024) achieve accurate global illumination via ray marching and neural fields, albeit computationally intensive. 3DGS (Kerbl et al., 2023) offers efficient scene representation by augmenting Gaussians with material properties (Liang et al., 2024; Gao et al., 2024a; Lai et al., 2025; Guo et al., 2024; Shi et al., 2025; Sun et al., 2025; Chen et al., 2025; Yao et al., 2025). For instance, GS-ROR[2] (Zhu et al., 2025) employs an SDF for geometric regularization and deferred splatting for faster rendering, though its reliance on split-sum approximation reduces accuracy in estimating materials and lighting. IRGS (Gu et al., 2025a) incorporates the full rendering equation in a 2D Gaussian ray tracing framework to capture inter-reflections and complex light transport, but its exhaustive stratified sampling is computationally inefficient and underperforms on glossy surfaces. To enable high-fidelity and efficient rendering of deformable objects under varied illumination, our DR-GS innovatively integrates these approaches while strictly preserving a complete physically-based rendering equation.

## 3 METHOD

This section introduces DR-GS, a decoupled deformable and relightable GS framework for physically-plausible rendering of deformable objects under varying illumination. Our three-stage pipeline first establishes the foundations of 2D Gaussian splatting and Gaussian ray tracing (Sec. 3.1). The static reconstruction stage builds initial 2D Gaussians and mesh representations from multi-view imagery (Sec. 3.2). We then solve the rendering equation to decouple material attributes from illumination while optimizing geometry (Sec. 3.3). Finally, we develop a physically-based deformation-driven method for 2D Gaussians using physical simulation and animation (Sec. 3.4). The full pipeline is illustrated in Fig. 2.

### 3.1 PRELIMINARY

**2D Gaussian splatting** (Huang et al., 2024) addresses key limitations of 3DGS (Kerbl et al., 2023), including its lack of explicit surface normals and multi-view inconsistencies, through geometrically constrained explicit surface representation. It compresses 3D Gaussians into 2D surfels parameterized by a center $p$, opacity $\alpha$, view-dependent color $c$, axial scaling $s = (s_u, s_v)$,

and a rotation matrix $\boldsymbol{R}$ built from orthonormal tangents $\boldsymbol{t}_u$ and $\boldsymbol{t}_v$. The surface normal is explicitly derived as $\boldsymbol{n} = \boldsymbol{t}_u \times \boldsymbol{t}_v$, ensuring view-consistent geometry. A central innovation is ray-splat intersection, which maps screen-space pixels to UV coordinates via the Gaussian kernel $G(\boldsymbol{u}) = \exp\left(-(u^2 + v^2)/2\right)$, augmented with perspective-correct splatting to reduce multi-view artifacts. Rendering uses depth-ordered alpha blending:

$$\boldsymbol{c}(\boldsymbol{r}) = \sum_{i=1}^{N} \boldsymbol{c}_i \alpha_i \hat{G}_i(\boldsymbol{u}) \prod_{j=1}^{i-1} \left(1 - \alpha_j \hat{G}_j(\boldsymbol{u})\right) \tag{1}$$

enabling end-to-end optimization of learnable parameters $\Theta_i = \{\boldsymbol{p}_i, \boldsymbol{s}_i, \boldsymbol{R}_i, \alpha_i, \boldsymbol{c}_i\}$.

**Gaussian ray tracing** integrates ray tracing techniques with Gaussian primitives to overcome limitations of rasterization-based Gaussian splatting in simulating effects like shadows and inter-reflections. The pioneering 3DGRT (Moenne-Loccoz et al., 2024) introduces particle-based ray tracing for 3D Gaussians, using a $k$-buffer hit-based marching algorithm with OptiX (Parker et al., 2010) hardware acceleration to improve both speed and accuracy. Following this advancement, 2DGRT (Gu et al., 2025a) resolves ray-splat inconsistencies through explicit surface representation and geometric constraints. This approach achieves physically accurate ray tracing, particularly for complex light paths involving multi-bounce indirect illumination.

## 3.2 STATIC RECONSTRUCTION

**Initialization.** Our framework begins by pretraining with Ref-Gaussian (Yao et al., 2025) to acquire initial 2D Gaussians $\Theta^P = \{\boldsymbol{p}, \boldsymbol{s}, \boldsymbol{R}, \alpha\}$ and subsequently extracts triangular mesh $\mathcal{M} = \{\mathcal{V}, \mathcal{F}\}$ via TSDF. DR-GS accommodates dual deformation approaches: particle-based deformation with optional mesh filling and mesh-based deformation necessitating reinitialization of $\Theta^P$ with $\mathcal{M}$.

**Mesh-based reinitialization.** We initialize a set of 2D Gaussians $\Theta^M = \{d, \boldsymbol{\omega}, \boldsymbol{s}, \boldsymbol{R}, \alpha\}$ on each triangular face $f \in \mathcal{F}$, where $\boldsymbol{R}$ aligns with the face normal $\boldsymbol{n}$, and $\boldsymbol{s}$ determined by local geometric properties. Same as GaussianMesh (Gao et al., 2024b), each Gaussian center $\boldsymbol{p}$ is parameterized by interpolation weights $\boldsymbol{\omega} = \{\omega_a, \omega_b, \omega_c\}$ and normal offset $d$, initialized as barycentric coordinates and zero, respectively. The relationship can be expressed as $\boldsymbol{p} = (\omega_a \boldsymbol{v}_a + \omega_b \boldsymbol{v}_b + \omega_c \boldsymbol{v}_c) + dr\boldsymbol{n}$, where $r$ denotes the circumradius of the associated triangle with vertices $\{\boldsymbol{v}_a, \boldsymbol{v}_b, \boldsymbol{v}_c\}$. Pretrained Gaussians $\Theta^P$ are projected onto mesh surfaces using spatial acceleration structure, retaining only interior projections. Finally, we compute averaged $\boldsymbol{\omega}$ and $\alpha$ per face and transfer them to corresponding mesh-based Gaussians $\Theta^M$, achieving geometry-aware reparameterization. See Alg. 1 for details.

## 3.3 PARAMETERS DECOUPLING

**Rasterization.** Following the same framework as IRGS (Gu et al., 2025a), DR-GS employs a physically-based deferred rendering pipeline: Gaussians are first rasterized to generate per-pixel maps, after which the rendering equation is applied. Each Gaussian is augmented with a set of material parameters $\Phi_{mat}$, which includes albedo $\boldsymbol{a} \in [0, 1]^3$, roughness $r \in [0, 1]$, and metallic $m \in [0, 1]$. Per-pixel attributes are aggregated via Gaussian rasterization:

$$\sum_{i=1}^{N} \gamma_i \{\boldsymbol{c}_i, d_i, \boldsymbol{n}_i, \boldsymbol{a}_i, r_i, m_i\}, \quad \text{where} \quad \gamma_i = \frac{T_i \alpha_i}{\sum_{k=1}^{N} T_k \alpha_k}, \quad T_i = \prod_{j=1}^{i-1}(1 - \alpha_j) \tag{2}$$

Here, $\boldsymbol{c}_i$ denotes the outgoing radiance, $d_i$ represents depth, and $\boldsymbol{n}_i$ is the normal vector. Leveraging the resulting depth map, the 3D surface point $\boldsymbol{x}$ corresponding to each pixel can be computed.

**Physically-based rendering.** To achieve photorealistic rendering, we employ the complete rendering equation (Kajiya, 1986), which is the fundamental formulation simulating light transport in physically-based rendering (PBR):

$$L_o(\boldsymbol{\omega}_o, \boldsymbol{x}) = \int_{\Omega} f(\boldsymbol{\omega}_o, \boldsymbol{\omega}_i, \boldsymbol{x}) L_i(\boldsymbol{\omega}_i, \boldsymbol{x})(\boldsymbol{\omega}_i \cdot \boldsymbol{n}) d\boldsymbol{\omega}_i \tag{3}$$

where $L_o$ and $L_i$ denote outgoing and incident radiance at point $\boldsymbol{x}$ in direction $\omega_o$ and $\omega_i$, respectively. The integral is computed over the hemisphere $\Omega$ around the surface normal $\boldsymbol{n}$. The bidirectional reflectance distribution function (BRDF) $f$ captures the material's light scattering behavior.

We further decompose incident light at the surface point $\boldsymbol{x}$ into direct and indirect components:

$$L_i(\boldsymbol{\omega}_i, \boldsymbol{x}) = V(\boldsymbol{\omega}_i, \boldsymbol{x})L_{\text{dir}}(\boldsymbol{\omega}_i) + L_{\text{ind}}(\boldsymbol{\omega}_i, \boldsymbol{x}) \tag{4}$$

where $L_{\text{dir}}$ represents distant illumination from an environment map, while visibility $V$ and indirect light $L_{\text{ind}}$ are calculated using 2DGRT (Gu et al., 2025a). Note that $L_{\text{ind}}$ is handled differently during reconstruction training and dynamic driving stages: during training, it is obtained by alpha-blending outgoing radiance $\boldsymbol{c}_i$ from Gaussians; the relight stage methodology is detailed in Sec. 3.4.

Given the incident radiance, we estimate the Eq. 3 via importance sampling (Cook & Torrance, 1982):

$$c_{\text{pbr}} = \frac{1}{N_r} \sum_{i=1}^{N_r} \frac{f(\boldsymbol{\omega}_o, \boldsymbol{\omega}_i, \boldsymbol{x})L_i(\boldsymbol{\omega}_i, \boldsymbol{x})(\boldsymbol{\omega}_i \cdot \boldsymbol{n})}{q(\boldsymbol{\omega}_i)} \tag{5}$$

where $N_r$ directions $\boldsymbol{\omega}_i$ are drawn from proposal distribution $q$ with PDF $q(\boldsymbol{\omega}_i)$.

**Training strategy.** Accurate modeling of light-surface interactions through geometry-aware ray tracing requires robust scene geometry. Precise disentanglement of geometry, illumination, and material is essential for flexible editing of virtual assets. We jointly optimize pretrained Gaussians $\Theta$ and material parameters $\Phi_{\text{mat}}$ while estimating illumination. To reduce computational cost, we selectively evaluate the Eq. 3 on a subset of pixels per view during training. The loss function is defined as:

$$\mathcal{L} = \mathcal{L}_c + \lambda_1^{\text{pbr}}\mathcal{L}_1^{\text{pbr}} + \lambda_{\text{light}}\mathcal{L}_{\text{light}} \tag{6}$$

where $\mathcal{L}_c$ denotes the RGB reconstruction loss from 3DGS (Kerbl et al., 2023) for rendered radiance $\mathcal{C}$, $\mathcal{L}_1^{\text{pbr}}$ represents the L$_1$ loss between physically-based rendered results and ground truth, and $\mathcal{L}_{\text{light}}$ regularizes incident illumination to natural white balance.

## 3.4 DYNAMIC DRIVING

**Particle-driven deformation**. Particle representations provide superior flexibility for modeling complex geometries. Conventional mesh extraction often yields non-manifold, self-intersecting, or non-watertight meshes (Shen et al., 2023), complicating physical simulations. High-resolution meshing incurs significant computational cost, while low-resolution alternatives lose geometric fidelity. We therefore employ particles for spatial discretization and leverage the MPM for simulation.

Specifically, we generate a particle cloud $\mathcal{P}$ by populating the interior of an extracted mesh $\mathcal{M}$ and incorporating its vertices. This hybrid representation supports subsequent mesh-based ray tracing and deformation-aware processing. Physical simulation is performed, yielding a deformed particle set (detailed in Alg. 2). We then compute the deformation gradient $\boldsymbol{F}_i$ and updated center position $\bar{\boldsymbol{p}}_i$ for each 2D Gaussian via GMLS interpolation (detailed in Alg. 3). The deformation gradient $\boldsymbol{F}_i$ is decomposed via polar decomposition into a rotation matrix $\bar{\boldsymbol{R}}_i$ and a scaling-shearing matrix $\bar{\boldsymbol{S}}_i$.

We efficiently apply $\bar{\boldsymbol{R}}_i$ and $\bar{\boldsymbol{S}}_i$ to each Gaussian as follows:

$$\boldsymbol{p}_i' = \bar{\boldsymbol{p}}_i, \quad \boldsymbol{R}_i' = \bar{\boldsymbol{R}}_i\boldsymbol{R}_i, \quad \boldsymbol{S}_i' = \text{diag}(\bar{\boldsymbol{\Lambda}}_i) \cdot \boldsymbol{S}_i \tag{7}$$

where

$$\bar{\boldsymbol{\Lambda}}_i = \begin{bmatrix} |\lambda_2(\bar{\boldsymbol{S}}_i)| & 0 \\ 0 & |\lambda_1(\bar{\boldsymbol{S}}_i)| \end{bmatrix}$$

**Mesh-driven deformation**. Mesh-based representations enable efficient editing, sculpting, animation, and relighting operations. For each deformed triangle $f' = (\boldsymbol{v}_a', \boldsymbol{v}_b', \boldsymbol{v}_c')$ in mesh $\mathcal{M}'$ and its corresponding face $f = (\boldsymbol{v}_a, \boldsymbol{v}_b, \boldsymbol{v}_c)$ in the canonical mesh $\mathcal{M}$, we compute a rotation matrix $\bar{\mathbf{R}}_i$, a shearing matrix $\bar{\mathbf{S}}_i$, and face-based displacement, which are directly applied to the bound Gaussians.

$$\begin{aligned} \Delta\mathbf{p} &= w_a(\boldsymbol{v}_a' - \boldsymbol{v}_a) + w_b(\boldsymbol{v}_b' - \boldsymbol{v}_b) + w_c(\boldsymbol{v}_c' - \boldsymbol{v}_c) \\ \bar{\mathbf{R}}_i &= w_a\bar{\mathbf{R}}_{v_a'} + w_b\bar{\mathbf{R}}_{v_b'} + w_c\bar{\mathbf{R}}_{v_c'} \\ \bar{\mathbf{S}}_i &= w_a\bar{\mathbf{S}}_{v_a'} + w_b\bar{\mathbf{S}}_{v_b'} + w_c\bar{\mathbf{S}}_{v_c'} \end{aligned} \tag{8}$$

The Gaussian center is updated as $\mathbf{p}_i' = \mathbf{p}_i + \Delta\mathbf{p}$, while the rotation and scale updates remain consistent with Eq. 7 and Eq. 8.

**Rendering acceleration and optimization**. During the dynamic driving phase, the original radiance values $c_i$ are no longer valid due to changes in the illumination conditions from those in the static reconstruction stage. We need to aggregate material attributes via Gaussian ray tracing and estimate incident radiance. To address the computational bottleneck of ray tracing, we replace 2DGS-based tracing with mesh-based ray tracing in this stage. Material properties learned through physical-based parameter decoupling (detailed in Sec. 3.3) are stored on a triangular mesh extracted via TSDF, enabling accelerated attribute lookup at the first ray intersection.

To further improve ray sampling efficiency, we employ multiple importance sampling (MIS) (Veach & Guibas, 1995), combining cosine-weighted, GGX, and environmental distributions to model diffuse, specular, and environmental lighting, respectively. The Monte Carlo estimator with the balance heuristic is defined as:

$$\sum_{i=1}^{n} \frac{1}{n_i} \sum_{j=1}^{n_i} w_i(X_{i,j}) \frac{g(X_{i,j})}{p_i(X_{i,j})}, \quad w_i(x) = \frac{n_i p_i(x)}{\sum_k n_k p_k(x)} \tag{9}$$

To mitigate inherent Monte Carlo noise, we incorporate a cross-bilateral filter based on spatiotemporal variance-guided filtering (SVGF) (Schied et al., 2017), which preserves geometric edges through depth- and normal-aware weighting.

## 4 EXPERIMENT

### 4.1 EXPERIMENT SETUP

**Datasets and metrics.** To evaluate the proposed method, we conducted experiments on the widely-used Glossy Synthetic dataset (Liu et al., 2023) and character models from SketchFab (Pinson, 2011) and Maximo (Adobe Inc., 2015) (including Vegeta, Mutant, NotEnrique). For quantitative assessment of static reconstruction quality, we employed three metrics for novel view synthesis: PSNR, SSIM (Wang et al., 2004), and LPIPS (Zhang et al., 2018).

**Baselines.** We adopt two categories of dynamic Gaussian Splatting baselines: particle-driven methods (*PhysGaussian* (Xie et al., 2024) and *GSP* (Feng et al., 2025)) and mesh-driven approaches (*SuGaR* (Guédon & Lepetit, 2024), *Mani-GS* (Gao et al., 2025), *GaussianMesh* (Gao et al., 2024b)). All baselines are re-implemented using the same input data for fair comparison. Implementation details are provided in Appendix E.

**Implementation details.** We first strictly adhere to the original configuration of Ref-Gaussian (Yao et al., 2025) for pre-training, followed by an extended fine-tuning phase of 20,000 iterations. For MIS, 512 rays are sampled during reconstruction (256 cosine-weighted, 128 GGX, and 128 light samples), while the number is reduced to 32 rays (16 cosine-weighted, 8 GGX, and 8 light samples) for dynamic driving and relighting stage. Environment maps are consistently set to a resolution of 128×256. All experiments are conducted on a single NVIDIA A6000 GPU.

### 4.2 EVALUATIONS AND COMPARISONS

#### 4.2.1 QUALITATIVE COMPARISON

Rendering results under geometric deformation are presented in Fig. 3, with key details emphasized through wireframe overlays. During static reconstruction, PhysGaussian erroneously bakes environmental reflections as static textures within its Gaussian representations, which prevents real-time reflection updates during object motion or deformation and compromises dynamic environmental interaction. Although GSP supports environment lighting decoupling, its poor surface reconstruction severely degrades performance on glossy objects.

Further comparisons in normal maps, rendering outputs, and estimated environment maps from static reconstruction are provided in Fig. 10, where GSP exhibits noticeable surface irregularities and hole artifacts. As observed in Fig. 4, surface unevenness in GSP becomes more pronounced after deformation, resulting in catastrophic relighting failures. In contrast, DR-GS produces smooth and coherent surface reconstructions while maintaining physically plausible appearances under simultaneous geometric and illumination changes. Detailed visualizations of the disentangled material parameters from DR-GS are provided in Fig. 11. Continuous deformation is shown in Fig. 12.

### 4.2.2 QUANTITATIVE COMPARISON

Table 1 presents a comparison of average quantitative metrics between ours and baseline approaches across all eight scenes of the Glossy Synthetic dataset. The results demonstrate that DR-GS outperforms all baselines in terms of PSNR, SSIM, and LPIPS. The PBR-based method GSP ranks second, underscoring the importance of physics-based modeling for reconstructing glossy objects.

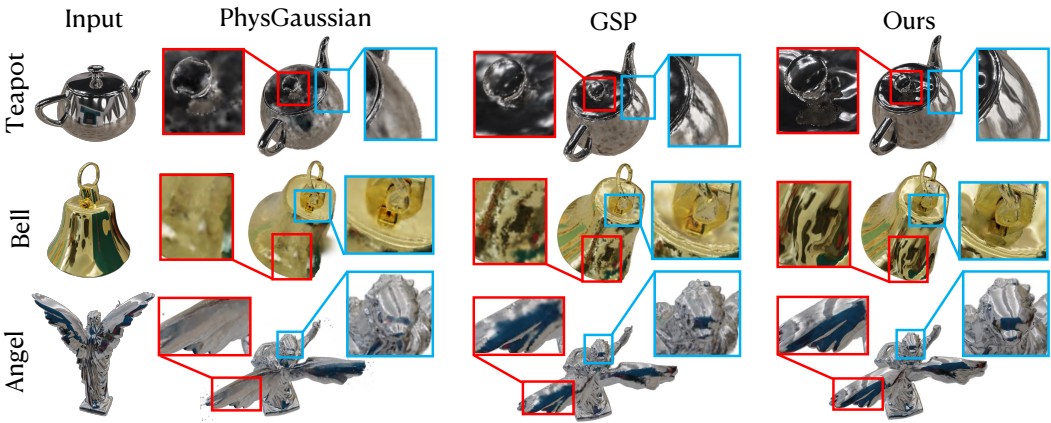

Figure 3: **Qualitative comparisons on deformed reflective scenes**. We present a comparative analysis of DR-GS against PhysGaussian (Xie et al., 2024) and GSP (Feng et al., 2025) in terms of geometric editing performance. The results demonstrate that ours achieves the clearest and most physically accurate reflection of the environment, even under large deformations of glossy objects.

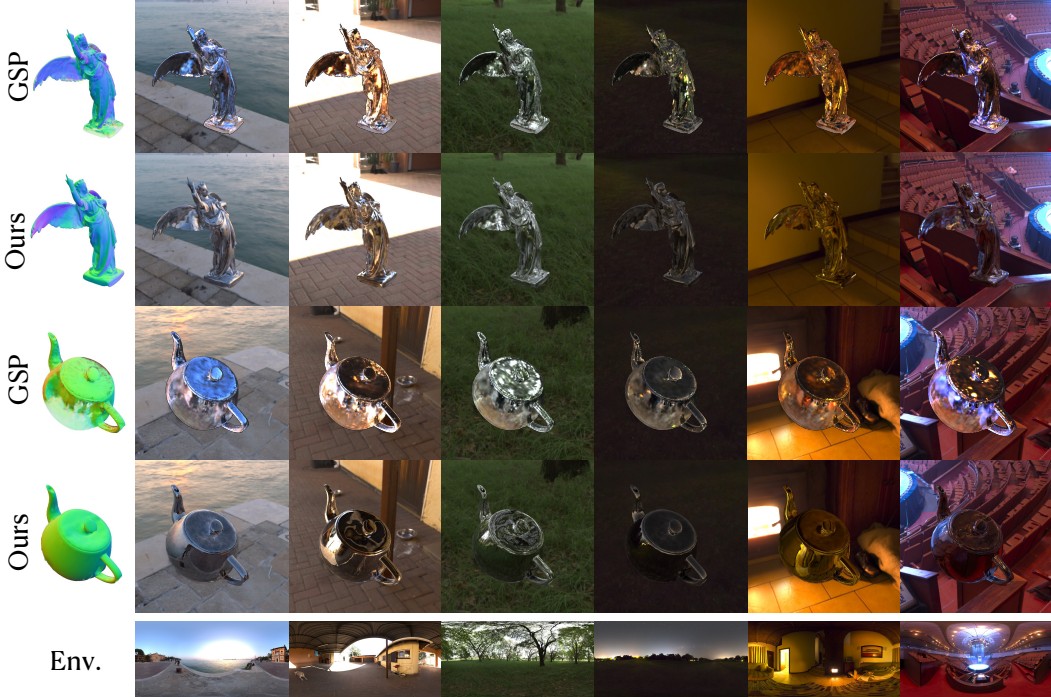

Figure 4: **Deformation results under relighting conditions**. We provide a visual comparison between our method and GSP (Feng et al., 2025) under combined geometric deformation and illumination changes. Results show that our method produces smoother normal maps and achieves physically plausible reflection under new lighting after deformation.

Table 1: **Quantitative comparisons of novel view synthesis on GlossySynthetic datasets** (Liu et al., 2023). The intensity of the red color signifies a better result. The distinction between **ours(P)** and **ours(M)** lies in the application of mesh-based reinitialization: the former operates directly without this step, whereas the latter incorporates TSDF-extracted mesh reinitialization. Detailed per-scene evaluation results are provided in Table 2.

|  | PhysGaussian | GSP | SuGaR | Mani-GS | GaussianMesh | **ours(P)** | **ours(M)** |
|---|---|---|---|---|---|---|---|
| PSNR↑ | 25.31 | 26.81 | 13.53 | 25.30 | 24.59 | **28.97** | 24.87 |
| SSIM↑ | 0.880 | 0.924 | 0.694 | 0.907 | 0.897 | **0.946** | 0.888 |
| LPIPS↓ | 0.110 | 0.091 | 0.246 | 0.096 | 0.101 | **0.065** | 0.114 |

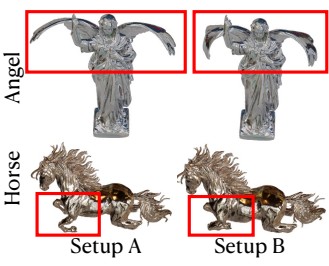

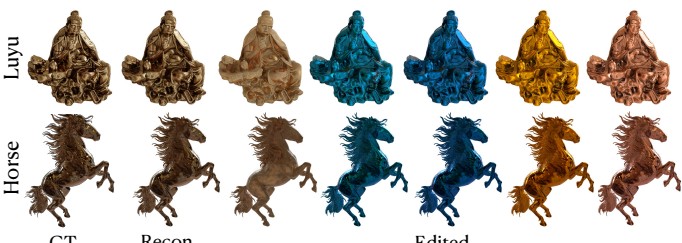

Figure 5: **Simulated deformation comparison**. The higher Young's modulus in Setup A diminished its elastic deformability compared to Setup B.

Figure 6: **PBR results with material parameter editing**. We present a comparison between DR-GS static reconstruction results and ground truth, along with rendered results after editing the decoupled material parameters.

## 4.3 PARAMETER EDITING

The fully decoupled parameter framework of DR-GS enables flexible post-reconstruction editing, facilitating versatile manipulation of virtual digital assets. Fig. 5 and Fig. 6 demonstrate edits of material properties related to simulation and rendering, respectively. Fig. 5 illustrates natural soft-body deformation under gravity, with red boxes highlighting the significant influence of parameter variations on the wings of Angel and the legs of Horse. In Fig. 6, composite editing of PBR parameters yields highly realistic and diverse rendering results.

## 4.4 MORE RESULTS

To assess the generalization of DR-GS, Fig. 3 showcases diverse geometric deformations and relighting outcomes under various illumination conditions. These include both bright and dim settings across indoor and outdoor environments, utilizing particle-driven and mesh-driven methods. Further results are available in the Appendix G and supplementary video.

## 4.5 ABLATIONS

We provide quantitative ablation studies for DR-GS below.

**Gaussian-based inter-reflection**. Reflective surfaces can generate indirect illumination through their own geometric structures, leading to inter-reflection effects. As shown in Fig. 8, DR-GS incorporates ray-traced visibility to synthesize plausible indirect illumination under novel views, achieving physically faithful modeling of inter-reflection.

**Robustness to extracted mesh**. Fig. 9 demonstrates that introducing learnable Gaussian attributes, i.e. normal offset $d$ and opacity $\alpha$, effectively enhances adaptability and robustness to inaccuracies in the extracted mesh. Without learnable $d$, the result exhibits noticeable spiky artifacts and structural blur; without learnable $\alpha$, scattered stain-like noise and localized crack-like distortions appear. In contrast, our full model achieves high-fidelity, clean, and detail-preserving rendering.

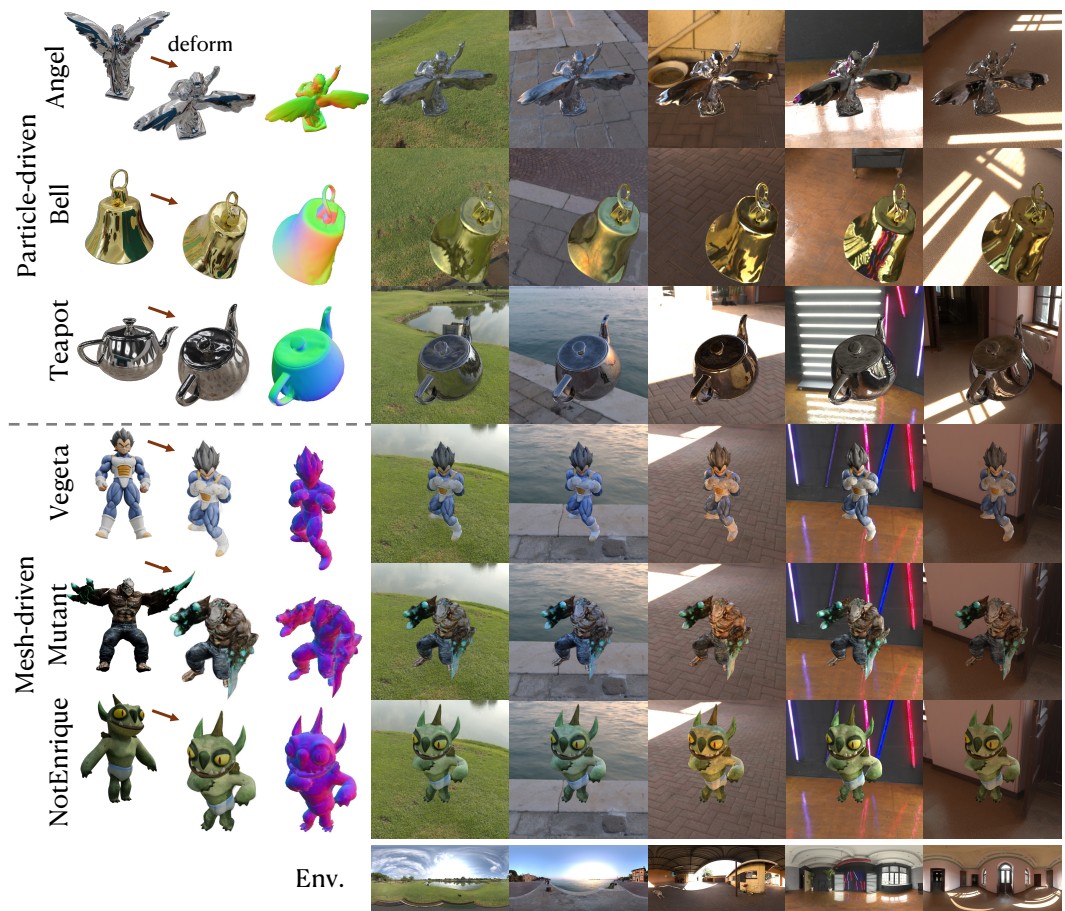

Figure 7: **More results**. It demonstrates reconstruction, geometric deformation, and physically-based rendering under various illumination conditions using estimated material parameters.

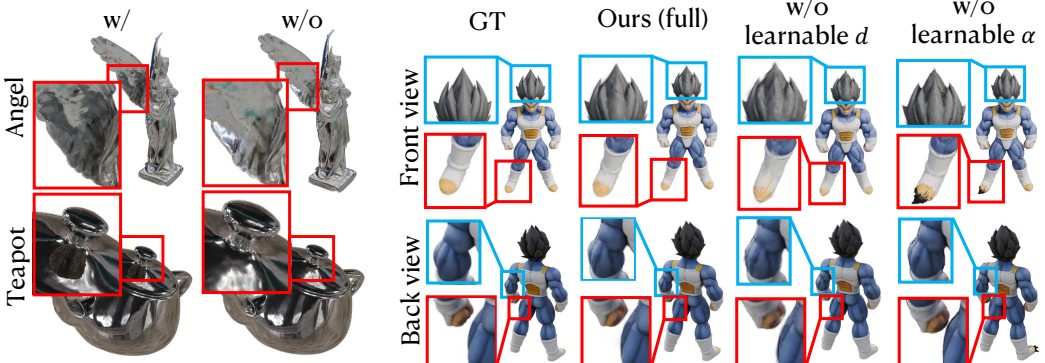

Figure 8: **Ablation on modeling of inter-reflection**.

Figure 9: **Ablation on learnable parameters**.

## 5 CONCLUSION

This paper presents DR-GS, a unified Gaussian framework for physically-based inverse rendering, relighting, and deformation-aware manipulation. By decoupling geometry, illumination, and materials, DR-GS addresses the unrealistic appearance of snapshots-based methods under geometric and lighting variations, while enabling flexible parameter control. We incorporate MIS and low-sample Monte Carlo estimation to achieve efficient dynamic rendering with high physical fidelity, particularly on glossy surfaces. Supporting both particle- and mesh-based deformation, DR-GS facilitates physical simulation and 3D animation, offering a new pathway for virtual content creation.

ETHICS STATEMENT

All datasets utilized in this study are sourced from publicly available repositories and do not contain any sensitive information.

REPRODUCIBILITY STATEMENT

The reproducibility of all experimental findings presented in this work is rigorously maintained. All data processing protocols and implementation codes will be publicly released upon article acceptance, supported by complete documentation to ensure full transparency and replicability of the research.

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

APPENDIX

## A USE OF LLMS

This research was conducted without the use of any large language models.

## B DETAILS OF MESH-BASED REINITIALIZATION

Our proposed mesh-based reparameterization approach is summarized in Alg. 1.

---

**Algorithm 1** Mesh-based 2D Gaussian Initialization

---

**Input:** Triangulated mesh $\mathcal{M} = \{\mathcal{V}, \mathcal{F}\}$, where $\mathcal{V} \in \mathbb{R}^{N_v \times 3}$ are vertex coordinates, $\mathcal{F} \in \mathbb{Z}^{N_f \times 3}$ are triangle face indices, pretrained 2D Gaussians $\Theta^P = \{\boldsymbol{p}_i, \boldsymbol{s}_i, \boldsymbol{R}_i, \alpha_i\}_{i=1}^{N_p}$.

**Output:** Initialized mesh-based 2D Gaussians $\Theta^M = \{d_j, \boldsymbol{\omega}_j, \boldsymbol{s}_j, \boldsymbol{R}_j, \alpha_j\}_{j=1}^{N_f}$

 1: **Initialize Mesh Gaussians $\Theta^M$:**
 2: **for** each face $f_j \in \mathcal{F}$ **do**
 3:       Compute face normal $\boldsymbol{n}_j$ via cross product: $(\boldsymbol{v}_{j2} - \boldsymbol{v}_{j1}) \times (\boldsymbol{v}_{j3} - \boldsymbol{v}_{j1})$
 4:       Initialize rotation $\boldsymbol{R}_j$ to align with $\boldsymbol{n}_j$             ▷ Quaternion from face normal
 5:       Set signed distance $d_j \leftarrow 0$
 6:       Compute face center $\boldsymbol{c}_j = \frac{1}{3}(\boldsymbol{v}_{j1} + \boldsymbol{v}_{j2} + \boldsymbol{v}_{j3})$
 7:       Compute nearest face distance: $\tilde{d}_j = \sqrt{\min_{k \neq j} \|\boldsymbol{c}_j - \boldsymbol{c}_k\|^2 + \epsilon}$    $(\epsilon = 10^{-7})$
 8:       Initialize logarithmic scale: $\boldsymbol{s}_j = \log(\tilde{d}_j) \cdot \mathbf{1}_2$
 9:       Initialize barycentric coordinates: $\boldsymbol{\omega}_j \leftarrow (\frac{1}{3}, \frac{1}{3}, \frac{1}{3})$
10: **end for**

11: **Project Pretrained Gaussians $\Theta^P$ to Mesh:**
12: Build BVH acceleration structure with $\mathcal{V}, \mathcal{F}$
13: **for** each Gaussian $\theta_i^P \in \Theta^P$ **do**
14:       $(d_i, f_i, \boldsymbol{\omega}_i) \leftarrow \text{BVH.signed\_distance}(\boldsymbol{p}_i)$                ▷ Find closest face
15:       **if** $\min(\boldsymbol{\omega}_i) > 10^{-3}$ **then**                ▷ Ensure projection is inside face
16:            $\mathcal{C}_{f_i} \leftarrow \mathcal{C}_{f_i} \cup \{(\boldsymbol{\omega}_i, \alpha_i)\}$                ▷ Add to candidate set
17:       **end if**
18: **end for**

19: **Aggregate Parameters:**
20: **for** each face $f_j$ with $\mathcal{C}_j \neq \emptyset$ **do**
21:       $\boldsymbol{\omega}_j \leftarrow \frac{1}{|\mathcal{C}_j|} \sum_{(\boldsymbol{\omega}, \alpha) \in \mathcal{C}_j} \boldsymbol{\omega}$
22:       $\alpha_j \leftarrow \frac{1}{|\mathcal{C}_j|} \sum_{(\boldsymbol{\omega}, \alpha) \in \mathcal{C}_j} \alpha$
23: **end for**

24: **Parameter Transfer:**
25: **for** each valid Gaussian $\theta_i^P$ with $f_i \geq 0$ and $\min(\boldsymbol{\omega}_i) > 10^{-3}$ **do**
26:       $\boldsymbol{\omega}_i \leftarrow \boldsymbol{\omega}_{f_i}$
27:       $\alpha_i \leftarrow \alpha_{f_i}$
28: **end for**

---

## C DETAILS OF PHYSICAL SIMULATION

Our proposed framework is theoretically compatible with any simulation methods. For the particle-driven experiments presented in Sec. 4, we employ the Material Point Method (MPM) (Hu et al., 2018), a hybrid numerical approach that synergizes the advantages of Lagrangian particles and Eulerian grids. MPM discretizes the continuum medium into material points, each point $i$ characterized by tracked position $\boldsymbol{x}_t(i)$, velocity $\boldsymbol{v}_t(i)$, and deformation gradients $\boldsymbol{F}_t(i)$ at timestep $t$. The momentum equations are subsequently solved on the Eulerian grid to achieve computational efficiency,

thereby naturally accommodating large deformation scenarios while circumventing the mesh distortion inherent in Lagrangian methods and the interface tracking challenges associated with Eulerian approaches. The detailed computational procedure is documented in Alg. 2.

---

**Algorithm 2** Material Point Method (MPM) Time Integration

---

**Input:** Particle positions $\boldsymbol{x}_t(i) \in \mathbb{R}^3$, velocities $\boldsymbol{v}_t(i) \in \mathbb{R}^3$, elastic deformation gradients $\boldsymbol{F}_t^e(i) \in \mathbb{R}^{3\times3}$ for $i = 1, \ldots, N_P$, time step $\Delta t \in \mathbb{R}^+$, external force field $\boldsymbol{b}(\boldsymbol{x}) \in \mathbb{R}^3$
**Output:** Updated positions $\boldsymbol{x}_{t+1}(i) \in \mathbb{R}^3$, velocities $\boldsymbol{v}_{t+1}(i) \in \mathbb{R}^3$, trial elastic deformation gradients $\boldsymbol{F}_{t+1}^{e,\text{trial}}(i) \in \mathbb{R}^{3\times3}$

1: **Initialize:**
2:    $\boldsymbol{v}_{t+1}(b) \leftarrow \boldsymbol{0}\ \forall b \in \{1, \ldots, N_G\}$
3:    $\boldsymbol{f}_t^\sigma(b) \leftarrow \boldsymbol{0}, \boldsymbol{f}_t^e(b) \leftarrow \boldsymbol{0}$
4: **for** each batch $\mathcal{B} \subseteq \{1, \ldots, N_G\}$ **do**
5:        $m_t(b) \leftarrow \sum_{i \in \mathcal{N}_b} N_b(\boldsymbol{x}_t(i))M(i)$ $\qquad\qquad\qquad\qquad\qquad$ ▷ Mass transfer
6:        $m_t(b)\boldsymbol{v}_t(b) \leftarrow \sum_{i \in \mathcal{N}_b} N_b(\boldsymbol{x}_t(i))M(i)\boldsymbol{v}_t(i)$ $\qquad\qquad\quad$ ▷ Momentum transfer
7:        $\boldsymbol{f}_t^\sigma(b) \leftarrow -\sum_{i \in \mathcal{N}_b} \frac{J(\boldsymbol{F}_t^e(i))}{\rho_0}\boldsymbol{\sigma}(\boldsymbol{F}_t^e(i))\nabla N_b(\boldsymbol{x}_t(i))M(i)$ $\qquad$ ▷ Internal force
8:        $\boldsymbol{f}_t^e(b) \leftarrow \sum_{i \in \mathcal{N}_b} \frac{J(\boldsymbol{F}_t^e(i))}{\rho_0}\boldsymbol{b}(\boldsymbol{x}_t(i))N_b(\boldsymbol{x}_t(i))M(i)$ $\qquad\quad$ ▷ External force
9:        $\boldsymbol{v}_{t+1}(b) \leftarrow \boldsymbol{v}_t(b) + \frac{\boldsymbol{f}_t^\sigma(b)+\boldsymbol{f}_t^e(b)}{m_t(b)}\Delta t$ $\qquad\qquad\qquad$ ▷ Grid velocity update
10: **end for**
11: **for** each particle $i \in \{1, \ldots, N_P\}$ **do**
12:        $\boldsymbol{v}_{t+1}(i) \leftarrow \sum_{b \in \mathcal{N}_i} N_b(\boldsymbol{x}_t(i))\boldsymbol{v}_{t+1}(b)$ $\qquad\qquad\qquad$ ▷ Velocity interpolation
13:        $\boldsymbol{F}_{t+1}^{e,\text{trial}}(i) \leftarrow (\boldsymbol{I} + \sum_{b \in \mathcal{N}_i} \boldsymbol{v}_{t+1}(b) \otimes \nabla N_b(\boldsymbol{x}_t(i))\Delta t)\boldsymbol{F}_t^e(i)$ $\quad$ ▷ Deformation update
14: **end for**

---

## D    DETAILS OF GMLS IMPLEMENTATION

Generalized moving least squares (GMLS) (Martin et al., 2010) represents a meshless generalization of Hermite interpolation in 3D space. Leveraging Alg. 3, we transfers motion information, including position $\boldsymbol{x}_i$ and deformation gradients $\boldsymbol{F}_i$, from simulation particles to target points corresponding to Gaussian center $\boldsymbol{y}_j$. The neighborhood relationship $\mathcal{N}_j$ between each target point $j$ and reference points $k$ is established through the precomputed binding matrix $\mathcal{B}_{jk}$ based on Euclidean distance, where subscripts $j$ and $k$ denote indices of target points and simulation particles respectively. Local weighting is achieved via weights $\boldsymbol{W}_j^k$ computed from the distance matrix $\mathcal{D}_{jk}$, while the choice of basis functions $\boldsymbol{\Phi}_k$ determines the order of interpolation accuracy. By constructing the regularized moment matrix $\boldsymbol{A}_j$ and solving the least quares problem to obtain coefficients $\boldsymbol{\alpha}_j$, the final interpolation of position and deformation gradients is computed at target point locations $\boldsymbol{\Phi}(\boldsymbol{0})$.

## E    DETAILS OF BASELINES

Our framework supports both particle-based dynamic driving paradigms, in contrast to most existing works that typically specialize in only one approach. For comprehensive evaluating, we selected mainstream methods representing each paradigm as baselines. To ensure a fair comparison, specific implementations of these baselines may differ from their original descriptions in order to maintain consistency with the input configurations used in our method. All models are trained for 30,000 iterations to achieve optimal performance.

### E.1    PARTICLE-DRIVEN BASELINES

---

**Algorithm 3** Generalized Moving Least Squares (GMLS) Interpolation

---

**Input:** Reference positions $\boldsymbol{X}_i \in \mathbb{R}^3$, deformed positions $\boldsymbol{x}_i \in \mathbb{R}^3$, deformation gradients $\boldsymbol{F}_i \in \mathbb{R}^{3\times3}$, target points $\boldsymbol{y}_j \in \mathbb{R}^3$, binding matrix $\mathcal{B}_{jk}$ (maps target $j$ to source $k$), distance matrix $\mathcal{D}_{jk}$, basis order $p \in \{1, 2\}$

**Output:** Interpolated positions $\boldsymbol{y}_j^{\text{interp}} \in \mathbb{R}^3$, interpolated deformation gradients $\boldsymbol{F}_j^{\text{interp}} \in \mathbb{R}^{3\times3}$

1: **Initialize:** $\boldsymbol{y}_j^{\text{interp}} \leftarrow \boldsymbol{0}$, $\boldsymbol{F}_j^{\text{interp}} \leftarrow \boldsymbol{I}$
2: **for** each batch $\mathcal{J} \subseteq \{1, \ldots, N_g\}$ **do**
3: $\quad \mathcal{N}_j \leftarrow \{k | \mathcal{B}_{jk} = 1\}, \forall j \in \mathcal{J}$ $\qquad\qquad\qquad$ ▷ Neighborhood selection
4: $\quad \boldsymbol{W}_j \leftarrow \exp(-\mathcal{D}_{jk}^2/\sigma^2), k \in \mathcal{N}_j$ $\qquad\qquad$ ▷ Weight computation
5: $\quad$ **if** $p = 1$ **then**
6: $\quad\quad \boldsymbol{\Phi}_k \leftarrow [1, \boldsymbol{X}_k^T]^T$ $\qquad\qquad\qquad\qquad\qquad$ ▷ Linear basis
7: $\quad$ **else**
8: $\quad\quad \boldsymbol{\Phi}_k \leftarrow [1, \boldsymbol{X}_k^T, (\boldsymbol{X}_k)_1(\boldsymbol{X}_k)_2, \ldots, \|\boldsymbol{X}_k\|^2]^T$ $\quad$ ▷ Quadratic basis
9: $\quad$ **end if**
10: $\quad \boldsymbol{A}_j \leftarrow \sum_{k\in\mathcal{N}_j} \boldsymbol{W}_j^k \boldsymbol{\Phi}_k \boldsymbol{\Phi}_k^T + \epsilon\boldsymbol{I}$ $\qquad\quad$ ▷ Regularized moment matrix
11: $\quad \boldsymbol{b}_j \leftarrow \sum_{k\in\mathcal{N}_j} \boldsymbol{W}_j^k \boldsymbol{\Phi}_k \boldsymbol{x}_k^T$
12: $\quad \boldsymbol{\alpha}_j \leftarrow \boldsymbol{A}_j^{-1} \boldsymbol{b}_j$ $\qquad\qquad\qquad\qquad\qquad$ ▷ Solve least squares
13: $\quad \boldsymbol{y}_j^{\text{interp}} \leftarrow \boldsymbol{\Phi}(\boldsymbol{0})^T \boldsymbol{\alpha}_j$
14: $\quad \boldsymbol{F}_j^{\text{interp}} \leftarrow \sum_{k\in\mathcal{N}_j} \boldsymbol{W}_j^k \boldsymbol{\Phi}_k \boldsymbol{F}_k \boldsymbol{A}_j^{-1} \boldsymbol{\Phi}(\boldsymbol{0})$
15: **end for**

---

- **PhysGaussian** (Xie et al., 2024) introduces a groundbreaking integration of Newtonian dynamics into 3D Gaussian Splatting by representing Gaussians as Lagrangian particles simulated within the Material Point Method (MPM) (Hu et al., 2018). This approach enables physically accurate animation of statically reconstructed Gaussian models through direct simulation, with kernel parameters updated according to the computed mechanical evolution.

- **GSP** (Feng et al., 2025) introduces a unified particle-based representation framework combining 3DGS and Position-Based Dynamics (PBD) (Müller et al., 2007), enabling physically consistent simulation of coupled solid and fluid dynamics within real-world reconstruction scenes. GSP decouples simulation particles from Gaussian kernels and drives the reconstructed model dynamically through interpolation of simulated attributes to each kernel.

For particle-driven approaches, we decouple the simulation particles from the Gaussian kernels and conduct physics simulations using an identical particle set. The resulting physical states are subsequently interpolated via GMLS to update each Gaussian kernel accordingly. Additional implementation details pertaining to the simulation setup and GMLS methodology are provided in Appendix C and D, respectively.

### E.2 MESH-DRIVEN BASELINES

- **SuGaR** (Guédon & Lepetit, 2024) introduces a regularization term that enforce precise alignment of Gaussian kernels with the underlying surface geometry, enabling rapid mesh extraction via Poisson reconstruction within minutes. By further proposing a Gaussian-mesh binding and co-optimization technique, the method not only enhances rendering quality but also supports direct mesh manipulation for editing Gaussian representations.

- **Mani-GS** (Gao et al., 2025) introduces a 3DGS editing framework based on triangular meshes. The core innovation lies in binding Gaussians to local triangle coordinates, enabling dynamic attribute updates driven by mesh deformation. Specifically, we initialize $N$=3 Gaussians per triangular face and optimize their attributes within local coordinates. During mesh editing, the system automatically updates corresponding Gaussians based on global coordinate transformations of the deformed triangles.

- **GaussianMesh** (Gao et al., 2024b) introduces a mesh-Gaussian coupling framework for real-time 3DGS deformation. The system establishes bijective mesh-Gaussian mappings and implements a dual splitting strategy that combines in-plane division with normal displacement. By propagating deformations via gradient fields, the framework synchronously updates Gaussian attributes during large-scale deformations, maintaining both geometric fidelity and rendering quality.

For mesh-based methods that rely on an explicit mesh representation, we supply the mesh obtained from the first stage of our method as input to both Mani-GS and GaussianMesh, as neither of their original publications specifies a concrete mesh extraction procedure. In the case of SuGaR, we adhere strictly to the framework outlined in its respective paper.

## F   DETAILS OF QUANTITIVE EVALUATION ON GLOSSY SYNTHETIC DATASET

Table 2: Per-scene static reconstruction quality comparison on GlossySynthetic (Liu et al., 2023) dataset. The intensity of the red color signifies a better result.

| Scenes | angel | | | bell | | | cat | | | horse | | |
|---|---|---|---|---|---|---|---|---|---|---|---|---|
| Metrics | PSNR↑ | SSIM↑ | LPIPS↓ | PSNR↑ | SSIM↑ | LPIPS↓ | PSNR↑ | SSIM↑ | LPIPS↓ | PSNR↑ | SSIM↑ | LPIPS↓ |
| PhysGaussian | 25.38 | 0.801 | 0.092 | 24.31 | 0.904 | 0.122 | 29.48 | 0.947 | 0.085 | 24.49 | 0.790 | 0.087 |
| GSP | 25.70 | 0.916 | 0.083 | 29.26 | 0.940 | 0.090 | 30.82 | 0.955 | 0.068 | 25.90 | 0.926 | 0.064 |
| SuGaR | 12.81 | 0.775 | 0.202 | 12.40 | 0.740 | 0.240 | 12.51 | 0.730 | 0.203 | 14.31 | 0.748 | 0.228 |
| Mani-GS | 26.39 | 0.911 | 0.075 | 24.58 | 0.901 | 0.114 | 29.90 | 0.950 | 0.072 | 24.76 | 0.897 | 0.075 |
| GaussianMesh | 25.84 | 0.901 | 0.079 | 24.02 | 0.890 | 0.118 | 28.39 | 0.936 | 0.083 | 23.70 | 0.886 | 0.081 |
| **ours(P)** | 29.64 | 0.943 | 0.052 | 31.94 | 0.962 | 0.056 | 31.25 | 0.965 | 0.052 | 25.54 | 0.929 | 0.055 |
| **ours(M)** | 25.04 | 0.887 | 0.094 | 28.12 | 0.917 | 0.103 | 27.07 | 0.931 | 0.091 | 21.25 | 0.851 | 0.111 |

| Scenes | luyu | | | potion | | | tbell | | | teapot | | |
|---|---|---|---|---|---|---|---|---|---|---|---|---|
| Metrics | PSNR↑ | SSIM↑ | LPIPS↓ | PSNR↑ | SSIM↑ | LPIPS↓ | PSNR↑ | SSIM↑ | LPIPS↓ | PSNR↑ | SSIM↑ | LPIPS↓ |
| PhysGaussian | 26.22 | 0.904 | 0.086 | 28.71 | 0.923 | 0.132 | 22.76 | 0.893 | 0.153 | 21.14 | 0.877 | 0.122 |
| GSP | 27.19 | 0.916 | 0.073 | 28.41 | 0.932 | 0.113 | 24.52 | 0.910 | 0.131 | 22.67 | 0.898 | 0.108 |
| SuGaR | 15.15 | 0.754 | 0.216 | 14.68 | 0.613 | 0.254 | 12.77 | 0.449 | 0.434 | 13.62 | 0.744 | 0.194 |
| Mani-GS | 26.42 | 0.909 | 0.069 | 27.83 | 0.922 | 0.110 | 21.38 | 0.885 | 0.145 | 21.18 | 0.878 | 0.106 |
| GaussianMesh | 26.02 | 0.905 | 0.071 | 27.53 | 0.918 | 0.112 | 20.55 | 0.871 | 0.156 | 20.64 | 0.871 | 0.109 |
| **ours(P)** | 28.57 | 0.940 | 0.053 | 30.55 | 0.941 | 0.095 | 27.54 | 0.942 | 0.094 | 26.72 | 0.945 | 0.062 |
| **ours(M)** | 24.10 | 0.873 | 0.098 | 26.75 | 0.887 | 0.145 | 23.67 | 0.879 | 0.165 | 22.93 | 0.882 | 0.107 |

## G   MORE RESULTS

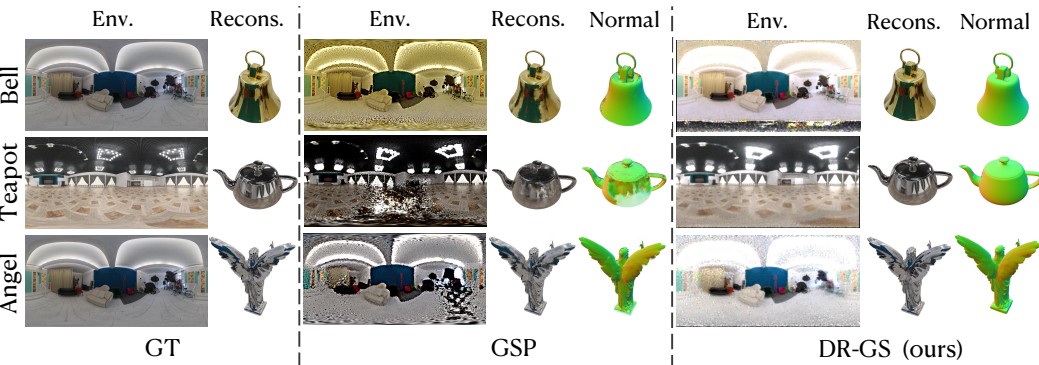

Figure 10: **Qualitative comparison of illumination decomposition and reconstruction**. It demonstrates a comparison between DR-GS and GSP in terms of environment lighting decoupling. Our method achieves clearer and more photorealistic novel view rendering results.

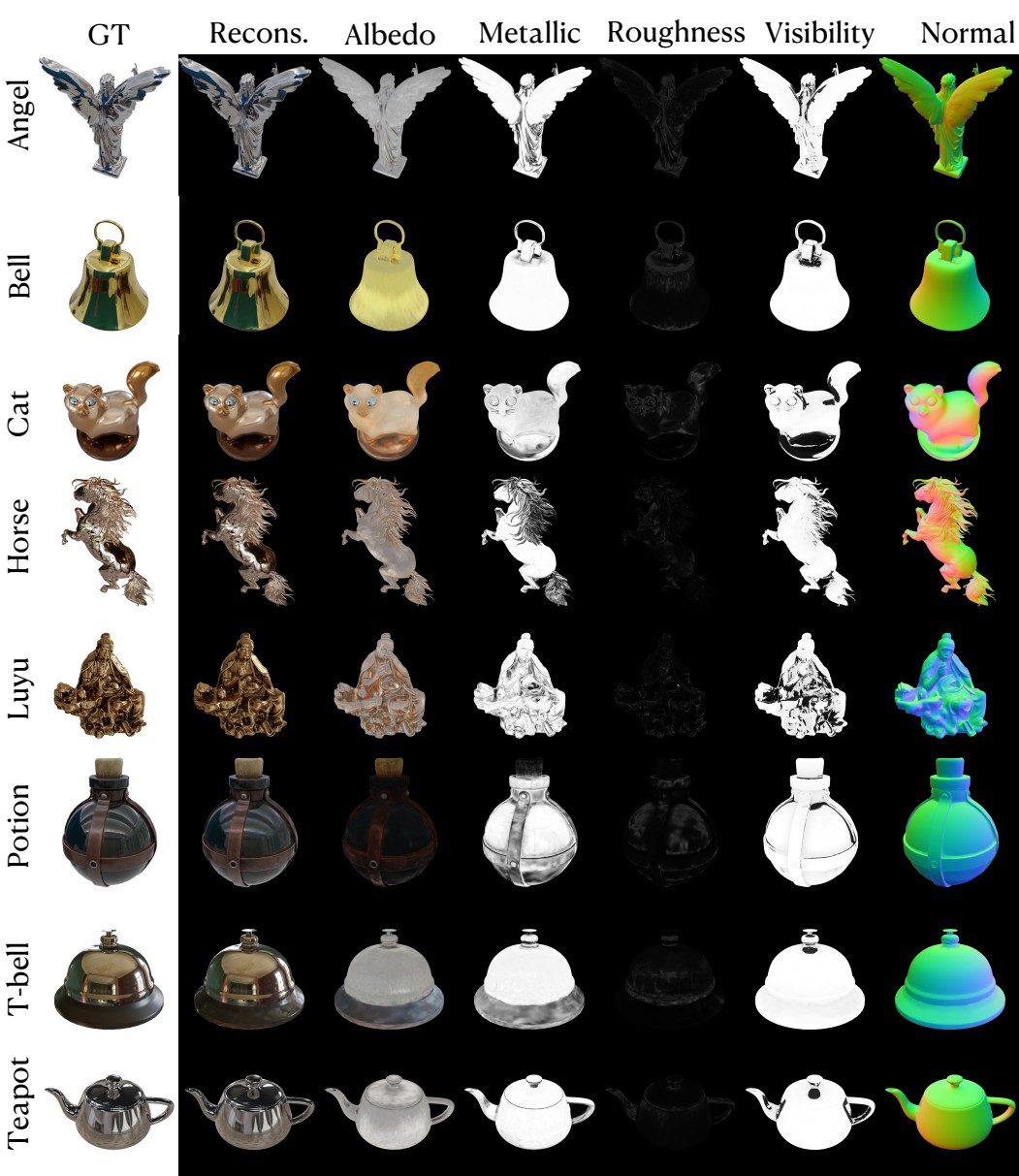

Figure 11: **Parameter disentanglement of DR-GS**. It presents reconstruction results in novel view, normal maps, visualizations of disentangled material parameters, and ray-traced visibility maps based on 2D Gaussians across all eight scenes of the GlossySynthetic dataset (Liu et al., 2023).

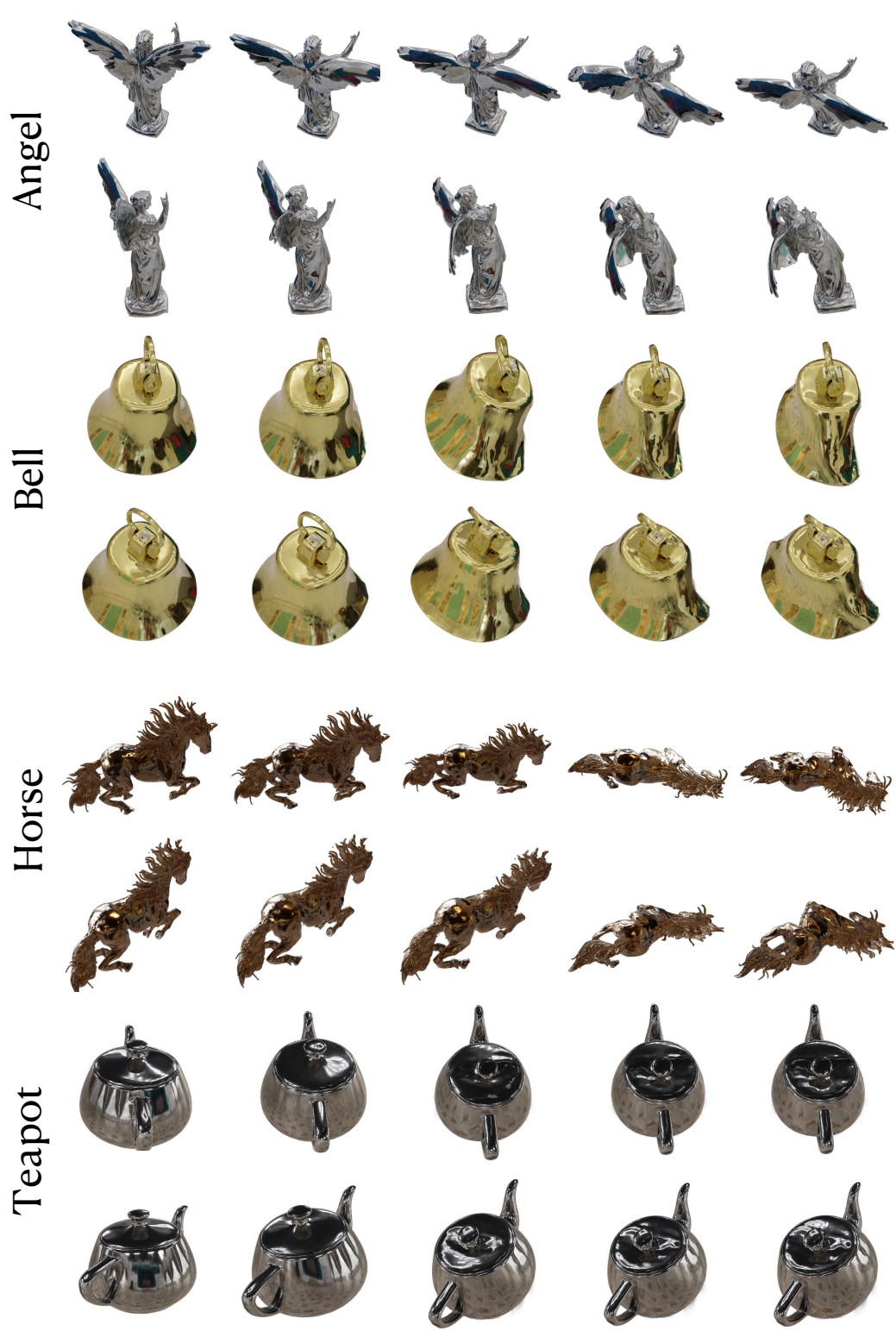

Figure 12: **Physical simulation results for glossy objects**. It presents continuous deformation sequences from soft-body simulations of objects in the GlossySynthetic dataset (Liu et al., 2023), captured from multiple viewing angles. The soft-body simulation generates diverse deformation states, demonstrating the capability of DR-GS to accurately reproduce environment lighting reflections under complex shape variations.

