# OpenReview forum: "Beyond Static Snapshots: Physically-Based Deformable and Relightable 2D Gaussians"
_ICLR.cc/2026/Conference — Submitted to ICLR 2026_

### Official Review · Reviewer_Qmmd · 2025-10-26

**Soundness:** 3
**Presentation:** 3
**Contribution:** 3
**Rating:** 4
**Confidence:** 3

**Summary:**

This paper introduces DR-GS, a framework that extends Gaussian Splatting (GS) to handle deformable objects with physically plausible rendering. This method tries to address limitations of existing GS methods, including baked-in illumination and restricted material editing, by decoupling geometry, illumination, and material representations. The framework integrates physically-based inverse rendering, relighting, and deformation-aware manipulation, enabling realistic rendering under dynamic conditions and post-reconstruction parameter editing. The paper demonstrates good visual fidelity in their results.

**Strengths:**

-They proposed a GS method for dynamic scenes with realistic illumination is an important and timely problem.

-The proposed framework with its decoupled representations and hybrid driving mechanism (particle and mesh) has not been seen in other works.

-The paper presents a complete system, including static reconstruction, parameter decoupling, and dynamic driving, demonstrating a strong engineering effort.

-The qualitative results show the framework's ability to handle deformations and relighting effectively, particularly on glossy surfaces where existing methods often struggle.

**Weaknesses:**

-Certain aspects of the method, particularly the GMLS interpolation for Gaussian updates and the specifics of how the hybrid driving mechanism works, are not described with sufficient clarity.

-They also lack justification of design choices. Some design choices, such as the particular selection of basis functions Φk in GMLS, lack justification. There are other choices which could have been explored.

-While the paper claims efficient dynamic rendering, it lacks runtime performance measurements. It would be useful to describe each stage runtime to understand where the bottleneck lies.

-The usage of particles are not fundamentally different with an explicit 3D deformation fields applied on the set of 3D points (or gaussian centres). More explanation and justification are needed to make it a solid contribution. Otherwise, it is just naming something known in a different way.

**Questions:**

-Can you provide a more detailed explanation of how GMLS is used to update Gaussian parameters based on particle/mesh deformations?

-The paper talks about different types of materials and their physical parameters. It would benefit from showing what realistic objects can be made.

- Can you elaborate on how the particle-based deformation idea is different with a deformation field applied on 3D points? For example, does the model can handle liquids or phenomenon like fire/fog, which are normally handled using particles?

---

### Official Review · Reviewer_zhch · 2025-10-29

**Soundness:** 2
**Presentation:** 2
**Contribution:** 2
**Rating:** 4
**Confidence:** 3

**Summary:**

This paper proposes DR-GS, an innovative unified deformable Gaussian framework that achieves the first efficient co-processing of inverse rendering, relighting, physics simulation, and 3D animation. The authors optimize rendering performance in dynamic scenes through multiple importance sampling (MIS) and Monte Carlo estimation, accelerating rendering while maintaining high visual realism. Meanwhile, the proposed pipeline also enables high-fidelity 3D asset creation and editing.

**Strengths:**

1. The authors propose a unified deformable Gaussian framework that handles deformation and relighting simultaneously.
2. The authors optimize rendering performance in dynamic scenes using MIS and Monte Carlo estimation, achieving both efficiency and high visual realism.
3. The decoupled geometry–illumination–material pipeline enables high-fidelity 3D asset creation and editing, advancing virtual content creation and simulation.

**Weaknesses:**

1. Could the authors add a discussion and comparison with Spring Gauss (Zhong et al., ECCV 2024)? This method also supports simulation and reconstruction simultaneously. For dynamic scenes, is it necessary to include deformable physical information in the 2D Gaussians?
2. The “deformable” part in this paper seems somewhat unclear. What is the exact purpose and novelty of the deformable component? Since the mesh is already extracted, mesh-driven deformation is a well-established technique, so there may be limited innovation in this aspect. The authors also claim that the framework supports both mesh-driven and particle-driven deformation — perhaps more motivation and justification for this unified design should be provided.
3. Although the authors emphasize a “unified framework” for dynamics and physics, could they provide more comprehensive comparisons with PhysGaussian? Additionally, some of the visual results presented in the paper are not very compelling, which raises doubts about the actual performance of the proposed method.

**Questions:**

Please refer to weakness

---

### Official Review · Reviewer_X4tV · 2025-10-29

**Soundness:** 3
**Presentation:** 3
**Contribution:** 3
**Rating:** 4
**Confidence:** 4

**Summary:**

This paper introduces DR-GS, a 2D Gaussian Splatting framework for synthesizing object deformation editing enhanced with physically-based rendering for updated rendering effects aligned with the object deformation. DR-GS integrates many recent advances in inverse rendering and deformation and achieves promising rendering results.

**Strengths:**

* This work achieves more accurate / photorealistic rendering effects (e.g., inter-reflections) for the post-reconstruction deformations.
* Efficient optimization for deformation rendering, e.g., proxy mesh for addition ray tracing, MC sampling strategies, etc.

**Weaknesses:**

* The method is well engineered, but the novelty is a bit limited. The main topic of this work is the post-reconstruction deformation rendering, but most components that contribute a lot to the rendering quality are from prior work: reconstruction and inverse rendering are mainly from Ref-Gaussian and IRGS; the MIS for MC ray tracing is also not new for rendering.
* The authors quantitatively compare the rendering quality against other baselines in Tab. 1 & 2. If I understand correctly, these metrics are evaluated before deformation. If all the baselines adopt the same Ref-Gaussian/IRGS reconstruction, the differences among these rendering metrics should be very small.
* The paper lacks quantitative metrics on
    * Deformation rendering quality;
    * Rendering efficiency: as this is one of the contributions of the work, further evaluation might be required.
* While the inter-reflection effects are well demonstrated in the paper, I don’t see any examples showing dynamic shadows during deformation. Correct shadow effects are also very critical for high-quality deformation rendering.

**Questions:**

As IRGS can help us get high-quality inverse rendering results, which can be easily converted to mesh objects with PBR textures, if we directly use the exported mesh for deformation with full ray tracing functionality, how will the results be?

---

### Official Review · Reviewer_Jw99 · 2025-10-30

**Soundness:** 2
**Presentation:** 3
**Contribution:** 2
**Rating:** 4
**Confidence:** 4

**Summary:**

This work proposes a novel framework for representing and rendering dynamic objects using 3DGS. It identifies two key limitations in existing dynamic 3DGS methods, i.e., (i) illumination is often baked into the texture, leading to incorrect appearance during deformation, and (ii) material properties are difficult to edit. To this end, the proposed method, DR-GS, incorporates the physically-based inverse rendering pipeline with deformation-driven 3DGS representations, delivering high-fidelity appearance for dynamic object reconstruction and enabling post-reconstruction material editing.

**Strengths:**

The paper is well-motivated, stemming from the critical observation that previous methods (PhysGaussian/GSP) often fail to deliver realistic results by estimating a baked-in appearance for dynamic objects. As a result, it is important to disentangle material properties from illumination, which enables physically plausible rendering by handling inter-reflections and self-occlusion as the object is animated. This can lead to more realistic results, especially for the challenging case of glossy objects, as shown in Figure 4.

**Weaknesses:**

### 1. Limited Technical Contribution
While the goal of creating a unified framework is compelling, the paper's core technical contribution appears to be an incremental integration of several existing techniques. The key components—inverse rendering for static Gaussians, physics-based simulation, and mesh-Gaussian binding—have been extensively explored in prior work.

- Inverse Rendering and Relighting: This has been the focus of numerous recent papers [1,2,3,4,5,6,7,8,9].
- Physics Simulation: Methods like PhysGaussian [10], VR-GS [11], and GSP [12] have already integrated physics into dynamic Gaussian Splatting.
- Mesh-Based Deformation: Binding Gaussians to a mesh for animation [13,14,15] and inverse rendering for raytracing [9] is also a well-established technique.

The paper does not sufficiently articulate the unique technical challenges that arise specifically from unifying these components. Without demonstrating and solving novel problems inherent to this integration, the contribution risks being seen as a skillful engineering effort rather than a fundamental advance.

Consequently, the claimed contributions feel overstated: the unified framework (L.085-L.087) is the main premise but lacks a clear innovative step, the dynamic rendering efficiency (L.087-L.089) is unvalidated, and the editing capability (L.089-L.090) is an expected benefit of material disentanglement, not a novel feature of this particular framework.

---

### 2. (Minor) Narrow Scope of Quantitative Evaluation
The paper's primary motivation is to improve rendering under dynamic deformations and illumination changes (L.086-L.087). However, the quantitative evaluation in Table 1 is conducted exclusively on the GlossySynthetic dataset [16], which consists entirely of highly reflective objects, making it difficult to assess whether the proposed techniques introduce a performance trade-off, potentially degrading quality on more common, diffuse scenes compared to simpler baselines. To support its broader claims, it is better to either present quantitative results on a more general dataset or scope its contributions more specifically to the relighting of deformable, reflective objects.

---

### 3. (Minor) Unvalidated Claims of Efficiency
The paper explicitly lists "Efficient dynamic rendering" as a contribution, stating it optimizes performance "through MIS and Monte Carlo estimation" (L.087-L.089). However, this claim is not substantiated with any empirical evidence. The experiments lack a quantitative analysis of rendering speed (e.g., FPS) or computational overhead compared to the baselines. Without this validation, the "efficiency" of the method remains an unproven assertion, and readers cannot assess the practical cost of achieving the high-quality visual results.

---

[1] GS-IR: 3D Gaussian Splatting for Inverse Rendering (CVPR'24)

[2] Relightable 3D Gaussian: Real-time Point Cloud Relighting with BRDF Decomposition and Ray Tracing (ECCV'24)

[3] SVG-IR: Spatially-Varying Gaussian Splatting for Inverse Rendering (CVPR'25)

[4] Reflective Gaussian Splatting (ICLR'25)

[5] IRGS: Inter-reflective Gaussian Splatting with 2D Gaussian Ray Tracing (CVPR'25)

[6] GI-GS: Global Illumination Decomposition on Gaussian Splatting for Inverse Rendering (ICLR'25)

[7] EnvGS: Modeling View-Dependent Appearance with Environment Gaussian (CVPR'25)

[8] GIR: 3D Gaussian Inverse Rendering for Relightable Scene Factorization (T-PAMI'25)

[9] GeoSplatting: Towards Geometry Guided Gaussian Splatting for Physically-based Inverse Rendering (ICCV'25)

[10] PhysGaussian: Physics-Integrated 3D Gaussians for Generative Dynamics (CVPR'24)

[11] VR-GS: A Physical Dynamics-Aware Interactive Gaussian Splatting System in Virtual Reality (SIGGRAPH'24)

[12] Gaussian Splashing: Unified Particles for Versatile Motion Synthesis and Rendering (CVPR'25)

[13] Mani-GS: Gaussian Splatting Manipulation with Triangular Mesh (CVPR'25)

[14] Real-time Large-scale Deformation of Gaussian Splatting (SIGGRAPH'24)

[15] MaGS: Reconstructing and Simulating Dynamic 3D Objects with Mesh-adsorbed Gaussian Splatting (ICCV'25)

[16] NeRO: Neural Geometry and BRDF Reconstruction of Reflective Objects from Multiview Images (SIGGRAPH'23)

**Questions:**

- The paper's mesh-based reinitialization (L.186-L.194) suggests the number of Gaussians is limited by the resolution of the extracted mesh. Based on my understanding, this could constrain the level of detail in the final appearance. Is this the reason for the slightly soft or less detailed results seen in the supplementary video? A clarification on the typical mesh resolution (face count) would be beneficial.

---

### Meta-Review · Area_Chair_Uu2h · 2026-01-07

**Summary:**

The reviewer raised the concerns regarding the novelty (with a list of relevant papers; raised by Jw99 , X4tV, zhch), and missing experiments for the claims (zhch, Jw99, Qmmd, X4tV).

We did not receive rebuttal for this paper.

As the reviewer's concerns are consistent and critical, AC's recommendation follows this trend.

**Reviewer Concerns:**

Unfortunately, there is no rebuttal thus it's hard to assess.

**Reviewer Scores:**

4 4 4 4 (original)

4 4 4 4 (after; no rebuttal)

---

### Decision · Program_Chairs · 2026-01-26

Reject